# The Development of Speaking and Singing in Infants May Play a Role in Genomics and Dementia in Humans

**DOI:** 10.3390/brainsci13081190

**Published:** 2023-08-11

**Authors:** Ebenezer N. Yamoah, Gabriela Pavlinkova, Bernd Fritzsch

**Affiliations:** 1Department of Physiology and Cell Biology, School of Medicine, University of Nevada, Reno, NV 89557, USA; enyamoah@gmail.com; 2Institute of Biotechnology CAS, 25250 Vestec, Czech Republic; gabriela.pavlinkova@ibt.cas.cz; 3Department of Neurological Sciences, University of Nebraska Medical Center, Omaha, NE 68198, USA

**Keywords:** hearing, auditory system, speaking, singing

## Abstract

The development of the central auditory system, including the auditory cortex and other areas involved in processing sound, is shaped by genetic and environmental factors, enabling infants to learn how to speak. Before explaining hearing in humans, a short overview of auditory dysfunction is provided. Environmental factors such as exposure to sound and language can impact the development and function of the auditory system sound processing, including discerning in speech perception, singing, and language processing. Infants can hear before birth, and sound exposure sculpts their developing auditory system structure and functions. Exposing infants to singing and speaking can support their auditory and language development. In aging humans, the hippocampus and auditory nuclear centers are affected by neurodegenerative diseases such as Alzheimer’s, resulting in memory and auditory processing difficulties. As the disease progresses, overt auditory nuclear center damage occurs, leading to problems in processing auditory information. In conclusion, combined memory and auditory processing difficulties significantly impact people’s ability to communicate and engage with their societal essence.

## 1. Introduction

Previous work initiated with Merzenich, Kaas, Mishkin, and Rauschecker demonstrated that the auditory cortex (AC), particularly in monkeys and humans, receives input from the medial geniculate body (MGB) and shows the best frequency along the lateral sulcus [1]. The auditory system is driven by sound sensing but is also affected by the interpretation of the acoustic environment. Various cortical neuron types can be related to the position in the AC [2]. Only about 18% of the AC, known as the primary AC (A1), receives input from the MGB. The function of the auditory system in acoustic information, such as sensing, processing, perception, and interpretation, is shaped by the convergence of subcortical and corticocortical information dispensation [3]. The AC is necessary for sound awareness and perception, detection, discrimination, categorization, and task-specific decision-making [2]. The AC is one node of an extended processing loop rather than a unidirectional information flow [4,5]. Our review mainly focused on how the developing infant’s cortex receives auditory input, which projects to the AC and extends beyond auditory nuclear centers to generate voice and singing, and how subsequent alterations reduce sound processing in seniors.

Speech, pitch, loudness, timbre, rate, phrasing, and non-verbal vocalizations convey human sounds and emotions. The ability to receive and interpret this information is unique to humans [1,6]. The evolution of spoken language must have emerged from animal neural mechanisms, and the capacity for vocalization is ancestral to tetrapods [7]. Various vocalizations include many mammals, birds, lizards, and frogs. Moreover, many fish, birds, and mammals generate sound in multiple ways [7,8]. Among the vocalized tetrapods, singing is distinct in humans. Singing may be human-specific, but ‘singing’ in whales constitutes sound variations subject to orchestral interpretations [9,10].

Moreover, the whales’ brain–body size ratio is similar to humans, providing an objective measure for the singing space [11]. Dolphins show alliances among each other using individually unique signature whistles. Whistles are learned early in life, allowing them to develop their whistle by making distinctions between those of other individuals [10]. In addition, singing in specific birds, like the nightingale, has distinct dynamic frequency shifts. Vocal production is known in songbirds, parrots, and hummingbirds, and vocalization in certain mammals (cetaceans, pinnipeds, elephants, bats, and humans) is commonplace. Like dolphins and humans, certain birds can learn different songs [12]. The human larynx and vocal tract are shared with other mammals. Incidentally, in humanoids, we lose a vocal sac in chimpanzees [7]. The avian syrinx is a unique source organ representing a novel way to sing in birds.

Nevertheless, the sound production mechanisms vary across mammals and birds [12,13]. Likewise, frogs produce sound with the larynx [14]. In contrast, bony fish generate sound in several ways [8]. Among the many sound productions, the most interesting is the siluriforms, which employ both sound and electric discharge, showing different communication strategies [15,16,17]. While land vertebrates use sound in the air, sound production somewhat differs depending on the hydrodynamic environment [18].

Interestingly, sound production may be controlled by similar motor neuronal input in fish, frogs, birds, and mammals [8,19]. The upstream of brainstem motor neurons is a direct input to activate sound production by higher orders, such as the cortex in mammals and certain sauropsids, including birds [12,19]. Picking out a relevant sound’s texture is critical for the auditory system. In particular, recognizing speech in noisy, singing, and whistling environments is challenging in developing infants and older people with auditory impairments [19,20]. The complete understanding of sound perception to sound emission requires at least an upstream from hair cells to the AC, which encompasses the Broca’s area and descends to the outflow to innervate the vagal output. 

Dyslexia is a polygenic developmental disorder characterized by a phonological and auditory deficit, often associated with changes in the micro-architecture in the temporal lobe [21,22]. Recent research indicates that specific hearing impairments are becoming correlated with highly heterogeneous genetics, with progress being made in identifying rare homozygous variants [23]. It appears that imbalances between excitation and inhibition in humans and mice might be the basis for auditory spectrum disorders. Among the monogenic models, the top priorities are Fragile X messenger ribonucleoprotein 1 (*Fmr1* [24,25]), SH3 and multiple ankyrin repeat domains 3 (*Shank3* [26]), phosphatase and tensin (*Pten* [27]), contacting associated protein-like 2 (*Cntnap2* [28]), among others. Future research necessitates the development of improved support for studying neuronal circuits with long-range hyperconnectivity and identifying the molecular players involved. These advancements will significantly contribute to a better understanding and definition of auditory dysfunction. Our review begins with the genetics of the auditory system of the AC, provides an overview of auditory dysfunction, describes the processes of learning speech and language, continues with discussing neurodegeneration in aging, and provides the relation between hearing loss and Alzheimer’s disease (AD) and the relation between the hippocampus and the auditory system.

## 2. Genes and Development of the Auditory Cortex

Evolutionary speaking, the forebrain consists of gene networks that are known across chordates [29,30,31] and active before neural tube closure of the neuropore [32]. Genetic factors that can play a role in dementia can have a distinction between familial vs. sporadic dementia and can also distinguish from polygenic risk scores that affect genomic research and treatment. Upstream is the earliest expression of paired box 6 (*Pax6*), T box brain transcription factor 1/2 (*Tbr1*/2), and forkhead box G1 (*Foxg1*; [30,33,34]). The *Wnt*/*β-catenin* is a signaling pathway that plays a crucial role in many developmental processes, including neuronal proliferation and differentiation of neural stem cells, by activating gene expression in cell cycle progression and neurogenesis [35,36]. The pathway involves a group of secreted glycoproteins (Wnts) that bind to the frizzled (*Fz*) receptor on the cell surface, leading to the activation of β-catenin that regulates the expression of various genes [36]. Dickkopf (DKK) is a negative regulator of the Wnt/β-catenin pathway. DKK proteins bind to and inhibit the function of a Wnt co-receptor, low-density lipoprotein-receptor-related protein 5/6 (*LRP5/6)*, which is necessary to activate β-catenin. Thus, DKK-1 inhibits the proliferation of neural stem cells and promotes their differentiation into mature neurons [37]. In addition, a set of genes is required for normal forebrain development [30,38,39,40]. The increase in the additional proliferation of the human brain is unclear since a few genes are involved [41]. Transcription factors vary precisely, inside–out, to create reproducible spatiotemporal patterns essential to gene regulatory networks in the developing forebrain [42,43,44,45,46,47]. The regional activation drives the production of region-specific cell types; for example, conditional cortex-specific paired box protein 6 (*Pax6*) deletion changes certain gene expressions, such as achaete-scute homolog1 (*Ascl1)*, neurogenin 2 (*Neurog2*), neurogenic differentiation 1 (*Neurod1*), *Tbr1*, SRY-related HMG-box 5/9 (*Sox5*, *Sox9*), and hairy and enhancer of split 5 (*Hes5*), while other genes are near normal, including (*Foxg1* [30]) and *Foxp2* [48,49]. Possible opposing interactions of *Pax6* and *Foxg1* may exist, both of which depend on the morphogens sonic hedgehog (*Shh)* and bone morphogenetic proteins (BMPs [47]). A loss of *Pax6* and *Foxg1* abolishes *Ascl1*, oligodendrocyte transcription factor 2 (*Olig2*), GS homeobox 2 (*Gsx2*), *Sox2*, and distal less homeobox-1/2/3 (*Dlx1*/*2*/*3)*, among others, which are redirected in a different neuronal variation. Downstream of *Pax6* is *Neurog2,* which regulates *Neurod1* expression. *Neurog2* and *Neurod1* heterodimerize to control progenitor cell production and the amplification of granule neuron progenitors, and the generation of the granule neurons of the hippocampus, which is dependent on *Neurod1* and lymphoid-enhancer-binding protein 1 (*Lef1* [50,51,52,53]). How many interneurons and primary pyramidal neurons develop remains unclear beyond the reduction of granule cells in *Neurod1* null mice.

In mice, neurons are born between embryonic day E11.5–13.5; in humans, it happens much later (5–6 gestational weeks; Figure 1). The earliest neurons are Cajal-Retzius neurons that develop slightly earlier in mice [E10.5–12.5; [50]]. Neuronal migration adds to the development of different brain layers [43]. Two regions are found in the preplate: the marginal and subplate zones (Figure 1; [54]). Early subplate neurons receive input from the thalamus and connect with cortical layer 4 neurons (Figure 1). The AC has six layers, as with most other cortex regions [55]. The maturation of connections from cortical neurons shows a delay in the somatosensory, visual, and AC regions, which undergo functional reorganizations to end up in oriented columns [56]. The delayed innervation of cortical layer 4, after an initial innervation from the subplate neurons, shows a typical progression in cortical excitatory ascending fibers (Figure 1). Most subplate neurons are lost in adults, and those that remain form layer 6. Surviving subplate neurons may support altered circuits, and such surviving neurons can cause neurological disorders [43]. Activation is provided by glutamatergic input to the subplate and layer 4, which will receive both AMPA and NMDA inputs and local GABAergic activity [54]. Progressive segregation from three closely derived thalamus connections (MGB, LGN, VPM) results in a discrete input/output relationship between the MGB and the AC in neonate mice [54,57]. 

A significant asymmetry between the left and right sides of the AC exists in humans, typically more prominent in the left hemisphere. In humans, the planum temporale, a cortical area crucial for auditory-language processing, is an asymmetric region up to ten times larger in the left cerebral hemisphere [58]. Similar asymmetry is also documented in chimpanzees, where the left side is more prominent. Bats, mice, gerbils, and rats also exhibit asymmetry, which may be involved in processing species-specific vocalizations [58,59]. The output from the cortex plays a crucial role in controlling the larynx and eventually reaches the vagal motoneurons, like other vertebrates. This neural pathway allows for precise and coordinated larynx movements, enabling various vocalizations and speech production in humans and other vertebrates [8,19].

## 3. Learning to Speak and Sing

Some sound vocalizations in humans can be activated without the need for learning. This phenomenon is exemplified by innate vocalizations, such as coughing, shared among amniotes [60]. Humans have learned a unique language [7,61,62], with similar associations in many vertebrates [19]. Before learning anything, we need to understand the origin of words: the infants’ early canonical babbling. Infants can produce speech-like sounds, such as consonant-vowel syllables, at about 6–12 months, which affects speech. The types of consonants and syllable structures most frequently used in vocalizations at the babbling stage are, for example, baba, mama, papa, nana, etc. [63]. Consolidating the infant babble requires a concise approach that helps understand the language’s development [64]. Once an infant understands words at about 12 months, the following steps can be built on the babble level. As toddlers grow, they rapidly expand their vocabulary at an impressive rate between 18 and 24 months. Normal infants aged 2–3 years can speak and sing [61]. The biggest question is: “Why study the origins of music, language, or any other human behavior? It’s unlikely that anyone will ever explain the full extent to which a particular behavior is accounted for by one or more adaptations because, given its complexity, human behavior cannot be exhaustively measured [65]”. Proto-musical behaviors could initially arise and spread as cultural inventions that had feedback effects on biological evolution, likely because of their impact on social bonding.

Two significant articles [61,65] somewhat contradict each other concerning music’s evolution. Both articles oppose that musical traits may be described as by-products of non-musical adaptive functions. In addition, both articles present unified theories aiming to explain the evolution of musicality in terms of music adaptation.

The marmoset monkey provides the best example of parent–infant vocal interaction. When the infant starts producing sound-like vocalizations, the parents will respond, shaping the infant’s vocalization [19] and providing feedback for social interaction. In addition, the continued vocabulary explosion may depend on association learning, familiar word recognition, and logically distinct associative learning [66,67].

Furthermore, singing and speaking to infants can positively impact their auditory development by exposing them to different sounds and patterns of speech, which can help them learn to process and understand language. Singing can be an engaging way for parents and caregivers to interact with infants and provide them with a rich auditory experience. Additionally, research has shown that singing can improve an infant’s ability to process speech sounds, potentially laying the foundation for later language development. Learning grammar through singing can help spelling and general learning [68].

Infants respond to the visual world early on, but it takes at least 1–2 years before vocal communication starts. Why does it take so long to speak when hearing starts before birth in the womb? How do these features change with infant age and differ across languages? The sounds the fetus hears in the womb are muffled and dampened due to the fluid surrounding them. However, studies have shown that even though the sounds are distorted, the fetus can still detect differences in pitch and volume [69]. Vocalizations may induce positive emotions through raised pitch and pitch variability and modulated loudness, reflecting speakers’ positive valence and heightened arousal [70]. Interestingly, even infant-directed songs in a foreign language can induce relaxation in babies, suggesting a common song effect worldwide. Mother–child interaction forms show differences between girls and boys and help them play synchronously [71].

The learning process in infants younger than one year remains somewhat unclear, despite the advancements in understanding infant-directed songs and words, which foster a dynamic and reciprocal interaction between the body, brain rhythms, and behavior. [72]. A newborn takes about 1–2 years to begin speaking and singing. After the first babble of words is uttered, the subsequent language development proceeds through continuous learning [66]. Of course, we know that older children can have language disorders, including attention deficit hyperactivity disorder (ADHD) and autism spectrum disorder (ASD; [73,74]), which may involve altered regulatory networks of genes relevant for language, including the transcription factor *FOXP2* [48].

## 4. Connections of the Auditory Cortex

Transient neurons and their connections formed during early brain development are transformed by spontaneous activity and by activity from the sensory periphery to mature cortical networks (Figure 1; [43]). The AC is a central hub located at a pivotal position within the auditory system, receiving tonotopic inputs from the MGB and playing a role in the sensation, perception, and interpretation of the acoustic environment [2]. ‘Receptive fields’ (RF) describe response properties of neurons activated with a particular stimulus dimension: the choice of acoustic stimulus dimensions is related to the primary perceptual attributes such as spectral pitch (frequency), loudness (intensity), periodicity, virtual pitch (amplitude modulations, harmonic series), timbre (spectral envelope), and sound location (interaural time, level differences, spectral shape). In humans, hemispheric asymmetries between the left and right AC are represented by gross anatomical features and cortical microcircuitry differences [1,75,76]. 

In the adult AC, two streams exit: the caudal belt reaching the dorsolateral prefrontal cortex, while the surrounding belt projects to the ventrolateral prefrontal cortex (Figure 2). Overall, the auditory ventral pathway plays a role in perception and is broadly consistent with a ‘what’ pathway, whereas the dorsal pathway has a sensorimotor function involved in the action (‘how’), including spatial analysis (Figure 2; [6]). Thus, speech processing of an anterolateral gradient is formed, in which the complexity of preferred stimuli increases, from tones and noise bursts to words and sentences, while a posterior-dorsal stream provides the spatial role and motion detection. Multiple parallel input modules are advocated for the dual-stream model (Figure 2).

A detailed analysis of auditory cortical processing in 360 cortical areas and 15 auditory cortical regions of 171 human connectomes at seven teslas was conducted with diffusion tractography for imaging and functional connectivity [77]. A hierarchy of auditory cortical processing was documented, which expanded from the AC to the belt regions, consistent with a ‘what’ ventral auditory stream, also known as language-related Broca’s area (area 45). Likewise, a ‘where’ stream from the superior parietal region may form a language-related dorsal stream to reach area 44 [77]. The A1 has substantial input from the MGB and connectivity from the visual (unidirectional) and somatosensory regions. Weak connectivity was shown for the cingulate area. Area 52 is unique to humans and receives substantial input from the MGB, weaker than the AC, which also connects somatosensory and insular regions. The belt input is reciprocal to the A1. In the auditory cortical processing hierarchy, the A1 is the strongest, followed by the belts and 52, A4, and finally A5 [77]. Information flow from areas 45 and 44 can be part of a route from semantic to language output in speech production and articulation [6].

In summary, human cortical connections are formed between the thalamus and the AC, extending through multiple steps of belt regions and reaching areas 52, 45, and 44. Future investigations may deepen our understanding of neural circuitry and show how various brain regions collaborate to support complex cognitive functions.

## 5. The Auditory Cortex Reaches Out

There are two major classes of neurons in the neocortex: principal pyramidal cells and interneurons. The AC is organized into six horizontal layers with anatomical and functional vertical columns and intense interhemispheric connections between the auditory cortices of both hemispheres [2,75]. Primary sensory cortices like A1, S1 (somatosensory cortex), and V1 (visual cortex) are not unimodal but can process other sensory information. Projections from different inputs arise from subgranular layers and provide feedforward organizations [75].

The auditory information is processed via the cochlear nuclei, superior olivary complex, lateral lemniscus, and inferior colliculi to reach the MGB [78,79]. The auditory pathways are formed through the combined contact of prominent inputs, including corticothalamic, cortico-collicular, colliculofugal, and olivocochlear connections [75,78,80]. A feedback loop modulates auditory response properties in the midbrain and hindbrain to alter their sensitivity to sound frequency, intensity, and location [75,78].

The auditory soundscape or the visual landscape can influence the perception in a natural, multisensory environment [81,82], from which comes visual and auditory input [77]. An interesting McGurke effect [83] provides interaction between visual and auditory stimuli [84]: When a video image of a mouth saying ‘g’ is played synchronously with playback of the sound ‘b,’ what is perceived is ‘d’, a sound intermediate in articulation. No study investigating this specific effect is known in animals, but recent research on audiovisual interactions in macaques suggests that monkeys also spontaneously link the auditory and visual components of conspecific calls, preferentially looking at video displays whose mouth shape matches a played call [7]. Multisensory interactions are audiovisual speech perception, in which visual speech substantially enhances auditory speech processes [85], more than what would have been expected from the summation of the audio and visual speech responses [86].

In addition to the direct input from the A1, higher-level connections between various brain regions influence and combine auditory sensations with inputs from other sensory systems. The auditory system is interconnected with the visual, somatosensory, taste, and vestibular systems, facilitating multisensory integration and enabling complex information processing [87].

## 6. The Role of the Hippocampus in Maintaining Hearing

Aging can cause a decline in various auditory system functions, including hearing sensitivity, speech perception, and the ability to understand speech in noisy environments. This decline can be caused by various factors, including changes in the inner ear, alterations in central auditory processing, exposure to noise, and other factors that can damage the auditory system over time [88]. However, the rate and extent of decline can vary significantly among individuals, and some degree of decline is a normal part of the aging process. Regular hearing screenings, consistent use of hearing protection, and maintaining a healthy lifestyle can effectively mitigate the impact of aging on the auditory system. As individuals age, there is a progressive loss of auditory input, beginning with the outer hair cells and extending to the inner hair cells. This is followed by the loss of sensory neurons in the cochlea [88]. By 2050, around 2.5 billion people worldwide will have hearing impairment [89]. Recent research has suggested that there may be a link between AD and hearing loss [88,89,90,91]. It is believed that the cognitive decline associated with the disease may affect the ability of the brain to process auditory information, leading to increased hearing loss. However, the use of hearing aids has been shown to improve hearing function in individuals with AD and may also help slow down the progression of the disease. The use of hearing aids can improve communication and social interactions, which can positively impact the overall quality of life for individuals with AD and hearing loss [90,91].

A novel idea of the effects of hearing aids on neurodegenerative diseases was further explored [92,93]. Griffiths et al. [92] grouped them into four potential mechanisms based on the typical cochlea, brainstem, and forebrain pathology. Hearing aids will help auditory hearing, but currently, there is not sufficient evidence to recommend the use of hearing aids to reduce cognitive decline [93]. The structural and functional features of the auditory brain could play a reciprocal interplay between peripheral and central hearing dysfunction [94], which seems to be particularly affected by the ‘tau’ proteins [95]. The initial stages of AD have been associated with dysfunction of the entorhinal cortex (Figure 3; [95]). Recent studies showed that enhancing the hyperexcitability of the cortical projection neurons in the lateral entorhinal cortex could facilitate the deposition of the amyloid β-protein and tauopathy in synaptically connected neurons in the hippocampus [96]. Currently, there is no cure for AD, and the available treatments can only moderately slow down the progression of the disease. While early diagnosis and treatment can help slow the disease’s progression and improve quality of life, they cannot cure it. Current treatment options focus on managing the symptoms and improving the individual’s ability to function and maintain independence for as long as possible [97]. It may involve a combination of medications, behavioral and psychological interventions, and support for the individual and their caregivers.

The hippocampus has three branches, CA1-CA3. In addition, the dentate gyrus receives the fibers labeled from the perforant path (green labeled with calretinin; Figure 3), whereas the granule cells (lilac) are positive for calbindin to differentiate into neurons that eventually reach out the nerve fibers [99,100]. The dentate gyrus of the hippocampus is one of the brain areas where neural stem cells persist during adulthood in most mammals [101]. Granule cells of the hippocampus depend on *Wnt*/*β-catenin*, *Dkk*, *Neurod1*, and *Lef1*, which will not develop if *Neurod1* and *Lef1* are deleted [35,37,51,53]. Labeling the newly formed granule neurons in the dentate gyrus showed that the maturation of adult-generated granule cells was slower than neonatal-generated granule cells [101,102,103]. These analyses suggest that the activity-dependent environment influences the maturation process of newly formed granule cells and their integration in the brain. Adult neurogenesis is characterized by lower rates of proliferation of neural stem cells that differ in their levels from quiescence to activation states [104]. The new formation of neurons appears in certain areas more and fewer in other areas in old mice and humans [100,101,105]. Adult hippocampus neurogenesis seems to be regulated by exercise, diet, and social interactions [93,103].

It is believed that the hippocampus helps to process sensory information, including auditory information, and integrates this information with existing knowledge and memories to create a coherent representation of the environment [92]. Studies have shown that the hippocampus is shaped by sound exposure and can be altered by changes in the auditory input [106]. In addition, recent research has also shown that the hippocampus plays a role in processing auditory information for speech perception and comprehension [98]. However, it is not directly involved in processing auditory information or multisensory integration. Instead, the hippocampus is primarily associated with consolidating short-term memories into long-term memories and spatial memory, which is the ability to navigate and remember spatial environments. These findings highlight the importance of the hippocampus in shaping our perceptions of the world and forming memories.

Further research is needed to understand better the specific mechanisms by which the hippocampus processes and integrates auditory information and how this information influences spatial navigation and memory formation [106] beyond the pleasure of music sound [107]. A hierarchy of sound can track how the hippocampus affects auditory information [98]. The evidence suggests a connection between the AC and the belt area reaching the perirhinal cortex and the hippocampus [77,98]. Correlating auditory experience can be identified with the extent of broad contributions, which, ultimately, connect hearing loss and dementia, including AD [98].

Pathological changes, including amyloid deposition (Aβ), neurofibrillary tangle (t-tau), and brain atrophy, present themselves years before dementia [95,108]. The association between hearing impairment, cognitive decline, brain structure, and Aβ and tau protein levels in the cerebrospinal fluid were investigated [109]. Poor hearing performance was associated with reduced amygdala, thalamus, and nucleus accumbens volumes and a high t-tau protein level. Hearing impairment was significantly associated with the volume of the hippocampus, but the association disappeared after the Bonferroni correction [109]. The significantly higher t-tau protein levels in the hearing impairment group require further research to establish the mechanisms underlying the link between t-tau protein and a volume reduction, particularly the temporal gyri [95]. 

In summary, the hippocampus is involved with hearing but is not critical, while tau levels strongly correlate with ADs and the temporal gyri. Further work is needed to consolidate AD’s relation to degenerative processes in the AC (a common cause of ADs affects AC) and how AC degeneration affects hearing.

## 7. Conclusions

Genetics play a role in the development of the auditory system, which allows infants to hear and process sounds. As they grow, they learn to speak and sing through exposure and practice, building upon their innate abilities. During the first 1.5 years of life, the auditory system rapidly develops, allowing infants to process and differentiate sounds, learn a language or more, and develop their speaking and singing abilities. However, their hearing abilities can decline as people age, leading to poor speech and language skills without proper care and support. The relationship between hearing loss and AD is still a topic of ongoing research and debate in the scientific community. The relationship between hearing loss and AD is likely complex and influenced by many factors, including genetics, lifestyle, and environmental factors. Further research is needed to understand the connection between hearing loss and AD and determine the best prevention and management methods.

## Figures and Tables

**Figure 1 brainsci-13-01190-f001:**
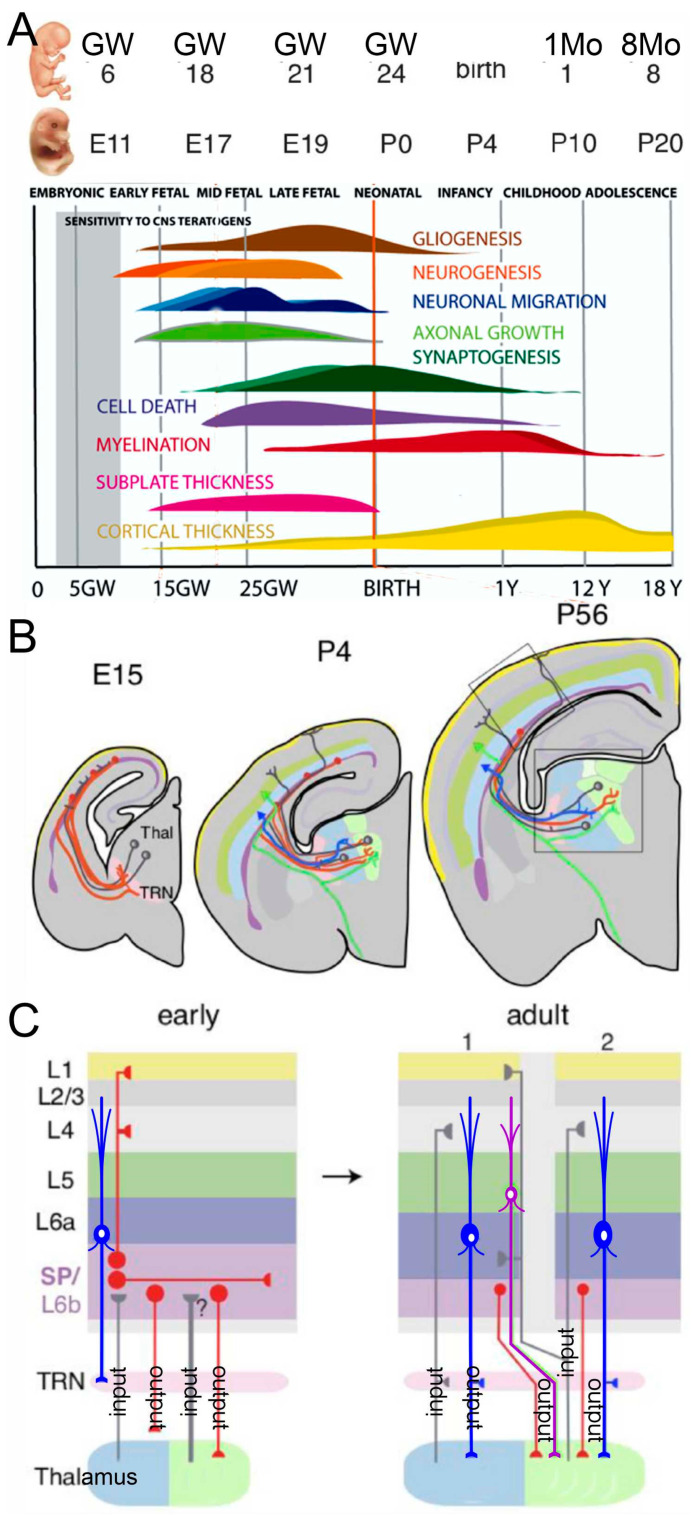
(**A**) This image shows the time points depicting the development of humans (in gestation weeks, GW; postnatal month, 1Mo; age, 1Y) and mice (embryonic, E; postnatal, P). Note that the first sound responses can be elicited in humans. Additional generation of granule cells is not indicated. (**B**) Thalamocortical connectivity is shown in mice that start in E15 and continue past P56, extending into the auditory cortex (AC). (**C**) The first reciprocal connections from the AC back to the thalamus (blue neurons in (**B**,**C**)) also have a transient input (TRN, early). Note that the adult mouse has a different input and output from primary (1) and secondary (2) input/output. It was modified after [43,56].

**Figure 2 brainsci-13-01190-f002:**
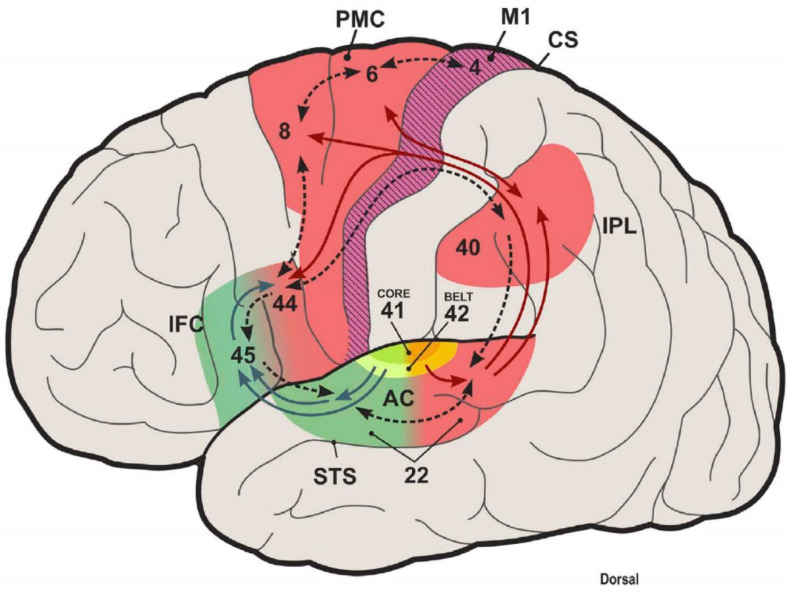
Dual auditory processing scheme of the human brain and the role of internal models in sensory systems. This scheme closes the loop between speech perception and production and proposes a typical computational structure for space processing and speech control in the posterodorsal auditory stream. Antero-ventral (green) and posterodorsal (red) streams originate from the auditory belt. The posterodorsal stream interfaces with premotor areas and pivots around the inferior parietal cortex. Here, predictive sensory information effect motor responses. A forward mapping is object information, such as speech, decoded in the anteroventral stream, including inferior frontal cortex (area 45) and motor-articulatory representations (area 44, ventral PMC), whose activation is transmitted to the IPL as an efference copy. An inverse mapping will attention- or intention-related changes in the IPL that influence the selection of context-dependent action programs in PFC and PMC. AC, auditory cortex; STS, superior temporal sulcus; IFC, inferior frontal cortex; PMC, premotor cortex; IPL, inferior parietal lobule; CS, central sulcus. Numbers correspond to Brodmann areas. Taken from [6].

**Figure 3 brainsci-13-01190-f003:**
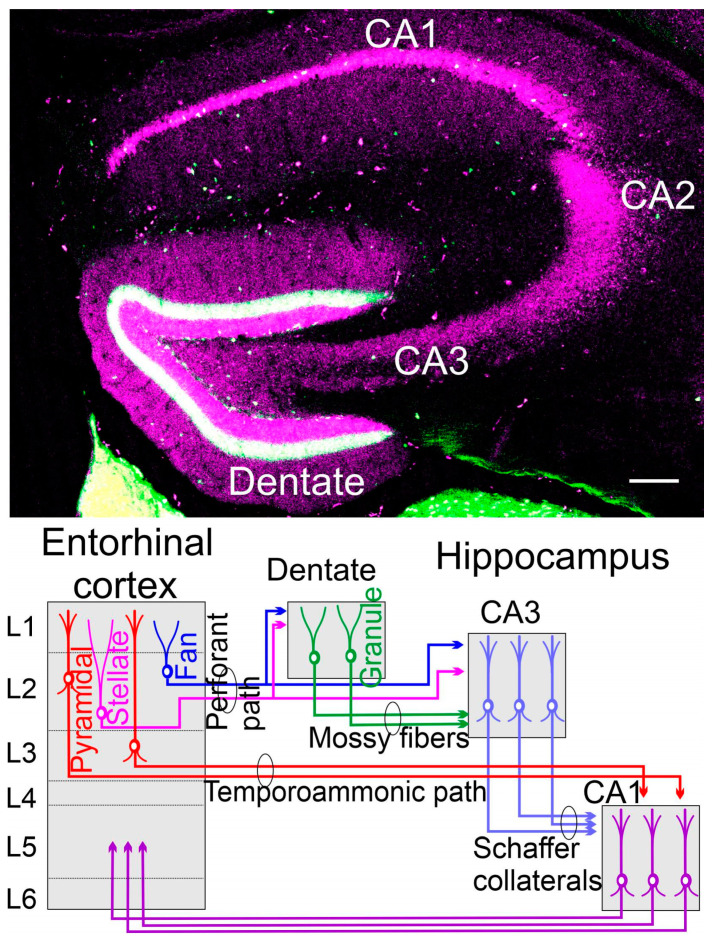
(**Top**) A coronal section is taken from a 48-day-old mouse using calbindin (lilac) and calretinin (green) in a transgenic expression. (**Bottom**) The dentate gyrus receives the perforant pathway from stellate and fan neurons that also split to innervate the CA3. Mossy fibers extend from granular neurons to innervate the CA3 by pyramidal neurons. Here are the Schaffer collaterals to innervate the CA1 that also receives the input from the temporoammonic path. The fibers’ process returns from CA1 to the entorhinal cortex in L5 by pyramidal neurons (not shown)—unpublished data (**top**) and modified (**bottom**) after [95,98].

## Data Availability

No new data were created or analyzed in this study. Data sharing does not apply to this article.

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
