# Peer review of "The Development of Speaking and Singing in Infants May Play a Role in Genomics and Dementia in Humans"

_brainsci, 2023, doi:10.3390/brainsci13081190_

Round 1
Reviewer 1 Report
General
The review covers various auditory topics which are very interesting: development of auditory cortex focused on molecular pathways involved, the role of music in speech and language development, the auditory system in dementia and the role of the hippocampus in hearing. All interesting but very diverse, and the authors do not link the processes in development to the processes in aging and dementia. The review reads like a summing up of various (quite random) facts about hearing. The second problem in the paper is that numerous sentences are grammatically incorrect and incomprehensible. Various examples are provided below.
In order to be acceptable for publication the authors need to choose one topic, and discuss that into depth in comprehensible English. It would involve a serious major revision.
Detailed comments
Title: with regard to “aging”, rather dementia and the role of hippocampus and memory is reviewed that aging. That should be reflected in title.
Line 27: A1 stands for primary auditory cortex (like V1 for primary visual cortex), not for auditory cortex as it is mentioned here. The authors may abbreviate auditory cortex with AC, and they should use A1 for primary AC.
Line 41: “main task”. It is not clear what is meant. Whose task?
Line 75-77. Example of sentence with language errors: “followed the role of learning of vocalization” and “the role of loses auditory system” and “the role of Alzheimer’s in the hippocampus”.
is meant: “then describe the processes of learning speech and language”, “continue with discussing neurodegeneration in aging”, and “the relation between hearing loss and Alzheimer’s and the relation between hippocampus and auditory system”?
Lines 81 and 89. Full names should be provided for the various genes.
Line 109-110. Which species does it relate to, the E10.5 -12.5? Mice, humans, or other?
Line 112-113. Example of incorrect and incomprehensible sentence. The reader has to guess what is meant. “Subplate is the first input from the thalamus”: does the subplate receive input from thalamus, or does the subplate provide input to the thalamus? and “...from the thalamus that switches to direct information of cortical layer 4”: does the thalamus provide input to layer 4 (as we know from the handbook), or is meant that the subplate provides input to layer 4?
Line 120. “Various activation” should be corrected
Line 122-124. Hard to read sentence.
Figure 1C. It would help to use arrows to indicate the direction of the connections, just like in Figure 3. The legend should explain the symbols used (filled circles, filled half circles).
Line 136. “Significant asymmetry between the hemispheres....typically more prominent in the left”: the second part of the sentence should be deleted (it is obvious that left and right differ when there is asymmetry) and instead the asymmetries should be specified. More grey or white matter L/R, more specialized for language L/R, more specialized for pitch L/R, and other aspects of sound. How large is the asymmetry?
Line 138. What is meant with “larger left side”, more grey or white matter volume?
Line 139-141. Sentence should be rewritten. “A correlation with human language processing..”: correlation of what? What is meant with the statement about similarities between humans and monkeys? Do monkeys process human speech as humans do?
Line 143-144. Unclear sentence.
Line 149. What is meant with “novel language”? Languages have evolved in human societies, so what is novel?
Line 151-152. Better to phrase it as “infants can produce speech-like sounds”. The authors often confuse the terms language and speech.
Line 164. What is meant with “music’s evolution”? That should be explained before it can be appreciated how this “evolution” can be contradicted.
Line 164-165. Better to phrase it as “both papers oppose the notion that musical traits may be described...”. The two theories should be explained in more depth. It is unclear what is opposed here, and what the alternative theory proposed in those two papers is.
Line 193-195. Unclear sentence. Rephrase and focus on what is known.
Line 196. “..first set of languages is uttered..” Here, the term “language” is incorrectly used. Words are uttered.
Line 197. Not clear why suddenly drug addictions are mentioned. It seems totally out of place.
Line 210. The sentence should be rewritten. Receptive fields describe response properties of neurons (or groups of neurons) as a function of certain stimulus characteristics such as frequency.
Line 214-216. See comment above about hemispheric asymmetries (line 136 and 138).
Line 257. Remove “that” and rewrite sentence to clarify what is meant here.
Line 268-270. The afferent auditory pathway should be described from peripheral to central.
Line 270-271. The auditory pathways (plural) should be specified.
Line 272-272. Sentence should be rewritten. “..from which …comes from” is not correct.
Line 286-288. The summary sentence is incomprehensible. For instance, “input of A1, which have higher connections….between connections” is incorrect English.
Line 297-298. Not clear how a hearing screening can minimize effect of age on hearing.
Line 299. “loses” should be “loss”, “starting from ….to..” should be “starting with....followed by”
Line 302-303. References should be provided as source of “recent research”.
Line 311-312. Four mechanisms are mentioned, which sounds quite interesting. It would be good to describe those four mechanisms, and to discuss them (which are the more likely ones?)
Line 330-331. “..neurons that reach out the nerve fibers”: not clear what is meant here, which neurons and which nerve?
Line 342-343. Interesting statement. The authors should elaborate on this.
Line 365. “..the pleasure of sounds”. What is meant here? Does it refer to music? And how does it relate to spatial navigation?
Line 365-366. Unclear sentence.
Line 367. “Various belts”. A more common term is “belt area”.
Line 371-372. Unclear sentence.
Line 372. What kind of cohort is referred to?
Line 374. Which volumes are meant?
Line 380-382. “hearing connections with the auditory cortex and Alzheimer’s disease”. I think the authors mean to say that the question is how Alzheimer’s relates to degenerative processes in the auditory cortex (common cause or Alzheimer’s affects AC) and how AC degeneration affects hearing.
Numerous sentences are grammatically incorrect and incomprehensible. Various examples are provided in the main comments.
Author Response
The review covers various auditory topics which are very interesting: development of auditory cortex focused on molecular pathways involved, the role of music in speech and language development, the auditory system in dementia and the role of the hippocampus in hearing. All interesting but very diverse, and the authors do not link the processes in development to the processes in aging and dementia. The review reads like a summing up of various (quite random) facts about hearing. The second problem in the paper is that numerous sentences are grammatically incorrect and incomprehensible. Various examples are provided below.
In order to be acceptable for publication the authors need to choose one topic, and discuss that into depth in comprehensible English. It would involve a serious major revision.
Detailed comments
Title: with regard to “aging”, rather dementia and the role of hippocampus and memory is reviewed that aging. That should be reflected in title.
Thank you, we have now changed the title: The development of speaking and singing in infants and the role of genomics and dementia in humans.
Line 27: A1 stands for primary auditory cortex (like V1 for primary visual cortex), not for auditory cortex as it is mentioned here. The authors may abbreviate auditory cortex with AC, and they should use A1 for primary AC.
Thank you for your suggestions, we start out with the auditory cortex (AC) and will later use A1 as the primary AC.
Line 41: “main task”. It is not clear what is meant. Whose task?
It was rewritten.
Line 75-77. Example of sentence with language errors: “followed the role of learning of vocalization” and “the role of loses auditory system” and “the role of Alzheimer’s in the hippocampus”.
is meant: “then describe the processes of learning speech and language”, “continue with discussing neurodegeneration in aging”, and “the relation between hearing loss and Alzheimer’s and the relation between hippocampus and auditory system”?
Thank you, we accepted these suggestions.
Lines 81 and 89. Full names should be provided for the various genes.
Indeed, we have now expanded the presentation by starting with the genes written in full name.
Line 109-110. Which species does it relate to, the E10.5 -12.5? Mice, humans, or other?
Just mice, human have a different trajectory (see Yamoah et al., 2020; for details). We included clear labeling of species.
Line 112-113. Example of incorrect and incomprehensible sentence. The reader has to guess what is meant. “Subplate is the first input from the thalamus”: does the subplate receive input from thalamus, or does the subplate provide input to the thalamus? and “...from the thalamus that switches to direct information of cortical layer 4”: does the thalamus provide input to layer 4 (as we know from the handbook), or is meant that the subplate provides input to layer 4?
We have corrected the text and provided Fig.1 which shows the different preplate and subplate, thank you for your help.
Line 120. “Various activation” should be corrected
Corrected, thank you.
Line 122-124. Hard to read sentence.
We have reworded the sentences, thank you.
Figure 1C. It would help to use arrows to indicate the direction of the connections, just like in Figure 3. The legend should explain the symbols used (filled circles, filled half circles).
Thank you, we highlight with ‘input’ and ‘output’ are now indicated..
Line 136. “Significant asymmetry between the hemispheres....typically more prominent in the left”: the second part of the sentence should be deleted (it is obvious that left and right differ when there is asymmetry) and instead the asymmetries should be specified. More grey or white matter L/R, more specialized for language L/R, more specialized for pitch L/R, and other aspects of sound. How large is the asymmetry?
The asymmetry is much larger on the left hemisphere in mice and humans (except for a few humans who have the right side larger).
Line 138. What is meant with “larger left side”, more grey or white matter volume?
The planum temporale area is much larger than the right side.
Line 139-141. Sentence should be rewritten. “A correlation with human language processing..”: correlation of what? What is meant with the statement about similarities between humans and monkeys? Do monkeys process human speech as humans do?
We have reworded the sentences: A significant asymmetry between the left and right sides of the auditory cortex is known to exist in humans, with this asymmetry typically being more prominent in the left hemisphere. In humans, the planum temporale, a cortical area crucial for auditory-language processing, is an asymmetric region that can be up to ten times larger in the left cerebral hemisphere [54]. Similar asymmetry is also documented in chimpanzees, where the left side is larger. Bats, mice, gerbils, and rats also exhibit asymmetry, which may be involved in processing species-specific vocalizations [54,55].
Line 143-144. Unclear sentence.
We have reworded the sentences: Some sound vocalizations in humans can be activated without the need for learning. This phenomenon is exemplified by innate vocalizations, such as coughing, which are shared among amniotes. [57].
Line 149. What is meant with “novel language”? Languages have evolved in human societies, so what is novel?
We have reworded the sentence.
Line 151-152. Better to phrase it as “infants can produce speech-like sounds”. The authors often confuse the terms language and speech.
Thank you, we have rephrased the sentences: Infants can produce speech-like sounds such as consonant-vowel syllables.
Line 164. What is meant with “music’s evolution”? That should be explained before it can be appreciated how this “evolution” can be contradicted.
The music evolution is well described in several papers. We have rephrased the sentences.
Line 164-165. Better to phrase it as “both papers oppose the notion that musical traits may be described...”. The two theories should be explained in more depth. It is unclear what is opposed here, and what the alternative theory proposed in those two papers is.
Thank you, the two theories are better presented.
Line 193-195. Unclear sentence. Rephrase and focus on what is known.
Reworded.
Line 196. “..first set of languages is uttered..” Here, the term “language” is incorrectly used. Words are uttered.
We have reworded line 196.
Line 197. Not clear why suddenly drug addictions are mentioned. It seems totally out of place.
We have taken out the statement, thank you.
Line 210. The sentence should be rewritten. Receptive fields describe response properties of neurons (or groups of neurons) as a function of certain stimulus characteristics such as frequency.
Reworded: ‘Receptive fields’ (RF) describe the response properties of neurons activated with a particular stimulus dimension
Line 214-216. See comment above about hemispheric asymmetries (line 136 and 138).
The planum temporale obviously shows the asymmetry in humans.
Line 257. Remove “that” and rewrite sentence to clarify what is meant here.
Reworded: In summary, cortical connections in humans are formed between the thalamus and the AC, extending through multiple steps consisting of belt regions, and further reaching area 52, 45 and area 44. Future investigations may deepen our understanding of neural circuitry and shed light on how various brain regions collaborate to support complex cognitive functions.
Line 268-270. The afferent auditory pathway should be described from peripheral to central.
Yes, reworded.
Line 270-271. The auditory pathways (plural) should be specified.
Reworded, thank you.
Line 272-272. Sentence should be rewritten. “..from which …comes from” is not correct.
Reworded:
Line 286-288. The summary sentence is incomprehensible. For instance, “input of A1, which have higher connections….between connections” is incorrect English.
Rephrased: In addition to the direct input from the A1, there are higher-level connections between various brain regions that influence and combine auditory sensations with inputs from other sensory systems. The auditory system is intricately interconnected with the visual, somatosensory, taste, and vestibular systems, facilitating multisensory integration and enabling complex information processing [83].
Line 297-298. Not clear how a hearing screening can minimize effect of age on hearing.
Reworded.
Line 299. “loses” should be “loss”, “starting from ….to..” should be “starting with....followed by”
Thank you, appreciated.
Line 302-303. References should be provided as source of “recent research”.
Thank you, added four citations.
Line 311-312. Four mechanisms are mentioned, which sounds quite interesting. It would be good to describe those four mechanisms, and to discuss them (which are the more likely ones?)
As stated before, currently there is not sufficient evidence to reduce cognitive decline.
Line 330-331. “..neurons that reach out the nerve fibers”: not clear what is meant here, which neurons and which nerve?
Reworded: whereas the granule cells (lilac) are positive for calbindin to differentiate into neurons that eventually reach out the nerve fibers
Line 342-343. Interesting statement. The authors should elaborate on this.
Added for clarification: However, it is not directly involved in processing auditory information or multisensory integration. Instead, the hippocampus is primarily associated with the consolidation of short-term memories into long-term memories and spatial memory, which is the ability to navigate and remember spatial environments. These
Line 365. “..the pleasure of sounds”. What is meant here? Does it refer to music? And how does it relate to spatial navigation?
Reworded, thank you
Line 365-366. Unclear sentence.
Reworded
Line 367. “Various belts”. A more common term is “belt area”.
Thank you, reworded.
Line 371-372. Unclear sentence.
Reworded, thank you
Line 372. What kind of cohort is referred to?
Reworded, thanks.
Line 380-382. “hearing connections with the auditory cortex and Alzheimer’s disease”. I think the authors mean to say that the question is how Alzheimer’s relates to degenerative processes in the auditory cortex (common cause or Alzheimer’s affects AC) and how AC degeneration affects hearing.
Thank you, reworded.
Reviewer 2 Report
Genomics is the study of all of a person's genes (the genome), including interactions of those genes with each other and with the person's environment. The topic is relevant and interesting. However, the title of this descriptive review does not fully correspond to the content. There is no list of the most promising candidate genes studied in clinical studies (involving humans) and genetic polymorphisms/variants that are associated with the development of speech and singing in humans.
Abstract and keywords need major correction, because it does not reflect the title of the manuscript, where "genomics" is in the first place.
The section "Introduction" needs modification, since the authors presented well-known encyclopedic knowledge about the function of the cortical auditory center in animals and humans, but they did not substantiate the significance and unresolved problems of genetic predisposition to speech development disorders in human ontogenesis.
Protein names should not be written in italics. I recommend the authors to correct this technical error in the text in all places where it occurs.
Please use italics only when writing the names of genes, but not proteins and signaling pathways.
Line 21 and further down the text - all abbreviations must be explained when they are first used. Please do not use abbreviations if you use them four or less times in this manuscript.
The title of the manuscript and the purpose of this descriptive review indicate that the authors want to update readers' knowledge about the genomics of speech development and its perception in humans. However, the authors discuss the studied genes, proteins and signaling pathways only in animals in the section "2. Genes and development of the auditory cortex" (in particular, in mice).
I suggest that the authors add tables with the results of genetic studies in humans (primarily) and animals (if they consider it necessary to leave the results of research on an animal model).
In sections 3-6, the authors did not disclose the role of genomics in general, although in Conclusion they write "Genetics plays a role in the development of the auditory system, which allows infants to hear and will process sounds.".
Minor correction of the English language style is necessary.
Author Response
Genomics is the study of all of a person's genes (the genome), including interactions of those genes with each other and with the person's environment. The topic is relevant and interesting. However, the title of this descriptive review does not fully correspond to the content. There is no list of the most promising candidate genes studied in clinical studies (involving humans) and genetic polymorphisms/variants that are associated with the development of speech and singing in humans.
Thank you for the positive review of our paper, it is appreciated. We have expanded candidate genes studied, in particular humans. Here is the new title: The development of speaking and singing in infants and the role of genomics and dementia in humans.
Abstract and keywords need major correction, because it does not reflect the title of the manuscript, where "genomics" is in the first place.
Thank you, we have now reconsidered our abstract and keywords.
The section "Introduction" needs modification, since the authors presented well-known encyclopedic knowledge about the function of the cortical auditory center in animals and humans, but they did not substantiate the significance and unresolved problems of genetic predisposition to speech development disorders in human ontogenesis.
We have expanded the introduction to provide speech disorders in human ontogenesis.
Protein names should not be written in italics. I recommend the authors to correct this technical error in the text in all places where it occurs.
Please use italics only when writing the names of genes, but not proteins and signaling pathways.
Thanks, proteins are not in italics.
Line 21 and further down the text - all abbreviations must be explained when they are first used. Please do not use abbreviations if you use them four or less times in this manuscript.
Thank you, all abbreviations are explained when they are first used.
The title of the manuscript and the purpose of this descriptive review indicate that the authors want to update readers' knowledge about the genomics of speech development and its perception in humans. However, the authors discuss the studied genes, proteins and signaling pathways only in animals in the section "2. Genes and development of the auditory cortex" (in particular, in mice).
We have slightly expanded Chapter 2 which details the auditory cortex, to include a description of a human’s. Here is a short summary of auditory dysfunction:
Dyslexia is a polygenic developmental disorder characterized by a phonological and auditory deficit, often associated with changes in the micro-architecture in the temporal lobe [21,22]. Recent research indicates that certain hearing impairments are becoming correlated with highly heterogeneous genetics, with progress being made in identifying rare homozygous variants [23]. It appears that imbalances between excitation and inhibition in humans and mice might be the basis for the auditory spectrum disorders. Among the monogenic models the top priorities are Fragile X messenger ribonucleoprotein 1 (Fmr1 [24,25]), SH3 and multiple ankyrin repeat domains 3 (Shank3 [26]), phosphatase and tensin (Pten [27]), contactin associated protein-like 2 (Cntnap2 [28]), among others. Future research necessitates the development of improved support for studying neuronal circuits with long-range hyperconnectivity and identifying the molecular players involved. These advancements will significantly contribute to a better understanding and definition of auditory dysfunction.
I suggest that the authors add tables with the results of genetic studies in humans (primarily) and animals (if they consider it necessary to leave the results of research on an animal model).
Thank you, instead we highlighted certain aspects associated with auditory dysfunction (see above).
In sections 3-6, the authors did not disclose the role of genomics in general, although in Conclusion they write "Genetics plays a role in the development of the auditory system, which allows infants to hear and will process sounds.".
Thank you, we have now presented the genetics for sections 3-6.
Comments on the Quality of English Language
Minor correction of the English language style is necessary.
Thank you, we have a reviewer who helped revise the language style with a native speaker.
Round 2
Reviewer 2 Report
I thank the authors for their answers and a small correction of the manuscript. However, the authors ignored the key questions about the genomics of the development and disorders of speech and singing in humans.
I have the impression that the title of the manuscript is misleading, so it needs to be stylistically improved.
Author Response
Thank you for your comments. We have changed the title:
The development of speaking and singing in infants may play a role in genomics and dementia in humans.
This captures the early development and relates it to elderly people.
Thank you for your suggestions.
Bernd Fritzsch, PhD